

# Measured and Modeled Snow Cover Properties across the Greenland Ice Sheet

Sascha Bellaire[1], Martin Proksch[1], Martin Schneebeli[1], Masashi Niwano[2], Konrad Steffen[3,4]

[1]WSL Institute for Snow and Avalanche Research SLF, Davos, Switzerland

[2] Meteorological Research Institute, Japan Meteorological Agency, Tsukuba, Japan

[3]Swiss Federal Institute for Forest, Snow and Landscape Research WSL, Birmensdorf, Switzerland

[4] Institute for Atmospheric and Climate Science, ETH Zurich, Switzerland

*Correspondence to: Sascha Bellaire (bellaire@slf.ch), Martin Schneebeli (schneebeli@slf.ch)*

**Abstract:** The Greenland ice sheet (GrIS) is known to be contributing to sea level rise in a warming climate. The snow cover on the ice sheet is the direct link between a potentially warmer atmosphere and the ice itself. However, little is known about the microstructure and especially about the spatial and temporal variability of the snow cover, except from indirect evidence from remote sensing. The detailed snowpack stratigraphy is relevant for processes such as the albedo feedback, water infiltration and firn densification. During a field campaign in 2015, spatially distributed snow observations of the GrIS were

gathered at stations belonging to the Greenland Climate Network (GC-Net). High-resolution snow profiles of density, specific surface area and hardness were measured. Hardness was measured with the SnowMicroPen, which was also used to assess the spatial variability of the snow density with depth. The snow cover model SNOWPACK was forced with reanalysis data from the model NHM-SMAP. The measured mean density of the upper snow cover was in good agreement with the simulations using constant densities for snow accumulation, i.e. new snow, depending on the geographical location on the GrIS. However,

the observed stratigraphy in terms of density and SSA could not be reproduced. We found that for a one-dimensional snowpack model it is difficult to parameterize for snowpacks undergoing multiple erosion and redeposition events, as is typical for the GrIS and other perennial polar snowpacks. This limitation may be a drawback to understanding past and future changes of the snow, and the associated processes.

Keywords: Greenland ice sheet, snow cover modeling, density, specific surface area, SNOWPACK, NHM-SMAP




## 1 Introduction

The Greenland ice sheet (GrIS) contributed around 8 mm to global sea level rise in the period from 1992 to 2011 (e.g. van den Broke et al., 2009; Shepard et al., 2012). The rate of melt and mass loss of the GrIS is increasing (Shepard et al., 2012, Orsi et al., 2017) and projected to further increase in the future (IPCC, 2014). Observed mass loss is largest at the coastal areas due to calving processes with a receding ice edge (e.g. Rignot and Kanagaratnam, 2006; van den Broeke, 2009). Recently, extreme melt events have been observed in the Greenland interior (e.g. Nghiem et al., 2012; Niwano et al., 2015), indicating that significant changes in the energy balance are now possible in the accumulation area of the GrIS. The fringe around the equilibrium line is the most affected area, with subsequent additional ice melt (Steger et al., 2017). Benson (1960) studied the stratigraphy of the snowpack in detail during several traverses. His study provided the emphasis for the work presented here, i.e. whether past warming affects metamorphism and snowpack stratigraphy. To this end, a snowpack simulation must be able to reproduce the current state, which is the topic of this article.

The snow cover in Greenland and its mechanical and structural properties are important for achieving a more detailed understanding of the firn densification and for improving larger-scale surface mass balance models (e.g. RACMO; van Meijgaard et al., 2008; MAR; Fettweis et al., 2007). Furthermore, in order to estimate the mass loss or gain due to melting or snow redistribution, the snow cover density and the change in specific surface area (SSA) are both required. Infiltration is largely affected by the stratigraphy and occurs mostly by preferential flow paths. The stratigraphy is therefore very relevant (Wever et al., 2015; Avanzi et al., 2017). An equally important factor is the albedo feedback, which is mainly a result of the rate of the reduction in SSA (Box et al., 2012).

Typically, due to the sparse availability of meteorological data, reanalysis data (e.g. Franco et al., 2012) or satellite products (e.g. Tedesco et al., 2012), or a combination of both (e.g. van Tricht et al., 2016), are used to model the GrIS snow cover and, hence, the surface energy and mass balance. The temporal resolution of these data is typically between 6 hours to 1 day, which implies that detailed changes in physical conditions of the GrIS snow are not fully captured.

Reanalysis products typically use simplified snow schemes, which lack the high resolution and degree of detail of snow cover models such as Crocus (Vionnet et al., 2012) or SNOWPACK (Bartelt and Lehning, 2002; Lehning et al., 2002a, b). Recently, van Tricht et al. (2016) used SNOWPACK to investigate the effect of clouds on the meltwater runoff on the GrIS. A detailed physical representation of the snow cover overlying the GrIS is in this respect indispensable to achieving a better understanding of the GrIS, in particular in a future warming climate.

Therefore, the main objectives of this study were modeling the present state of the GrIS overlaying snow cover in terms of density and stratigraphy. Simulations were carried out using the physically based snow cover model SNOWPACK forced with reanalysis data. These simulations were then compared with high-resolution snow cover profiles across the GrIS, which were obtained during a field campaign in 2015.



## 2 Data and Methods

### 2.1 NHM-SMAP

For this study, we used high-resolution (5 km) atmospheric reanalysis data generated by dynamically downscaling the Japanese Meteorological Agency's (JMA) latest reanalysis data, JRA-55 (the Japanese 55-year reanalysis; Kobayashi et al., 2015), with the JMA's operational nonhydrostatic atmospheric model, JMA-NHM (Saito et al., 2006), between 2010 and 2015. JMA-NHM has been used extensively for short-term weather forecasts on the GrIS to support field campaigns (Hashimoto et al., 2017), and its good performance on the GrIS has been demonstrated. Energy exchanges between the atmosphere and the snow/ice surfaces are calculated by a boundary-layer scheme implemented in the physical snow cover model SMAP (Niwano et al., 2012). The model domain in the present study covers the entire GrIS, and is the same as that documented by Hashimoto et al. (2017). This dataset will here on be referred to as NHM-SMAP.

Concerning the model setting of JMA-NHM, we followed Hashimoto et al. (2017). In addition to a single-moment scheme – which only predicts the mixing ratio of particles of liquid hydrometeors, i.e. cloud water and rain – a detailed double-moment bulk cloud microphysics scheme was used to predict both the mixing ratio and concentration of particles of solid hydrometeors, i.e. cloud ice, snow, and graupel. A cumulus parametrization scheme was not used.

The NHM-SMAP model calculations were conducted using the so-called 'weather forecast mode'. This means that atmospheric profiles were initiated daily at 18 UTC using JRA-55 and simulations with a lead time of 30 hours were performed accordingly. Considering the model spin-up, results for the first 6 hours were neglected. As a result, hourly atmospheric data from 00 to 23 UTC were created daily. The surface meteorological properties provided by NHM-SMAP include precipitation, incoming shortwave and longwave radiation, cloud fraction, surface pressure, 2 m air temperature, 2 m relative humidity – with respect to water – as well as the 2 m wind speed. The time series of meteorological NHM-SMAP data were extracted at stations of the Greenland Climate Network (GC-Net; Steffen and Box, 2001; Fig. 1) using the nearest-neighbor grid cell.

### 2.2 Greenland Climate Network

In order to assess the performance of NHM-SMAP, we compared the simulated air temperature, wind speed and incoming shortwave radiation with the measured (GC-Net) values. For comparison, we used hourly data from the Summit station between September 2010 and May 2015. Air temperature and wind speed were taken from the upper level sensors, namely the Type-E thermocouple for the air temperature and the RM Young 05103 for wind speed. Incoming shortwave radiation was measured with the LI-COR Photodiode. Further details on the GC-Net stations as well as data quality checks are provided by Steffen and Box (2001) and Box and Steffen (2000).





### 2.3 SNOWPACK

The snow cover model SNOWPACK (Bartelt et al., 2002; Lehning et al., 2002a, b; version 3.4.0) was forced with meteorological reanalysis data provided by NHM-SMAP. SNOWPACK can be driven using various combinations of input parameters. We chose to force SNOWPACK using incoming shortwave and longwave radiation, as well as liquid precipitation

along with air temperature, relative humidity and wind speed. Note that relative humidity in NHM-SMAP is calculated in relation to water and was corrected with respect to ice according to:

$$RH_{ice} = \frac{e_w}{e_i} \times RH_{water}, \tag{1}$$

with $RH_{ice}$ as the relative humidity over ice and $RH_{water}$ as the relative humidity over water. $e_w$ and $e_i$ denote the saturation vapour pressure over water and ice, respectively, and are defined as:

$$e_w = 611.2 \cdot e^{\frac{17.62 \cdot T}{243.12 + T}} \quad ; \quad e_i = 611.2 \cdot e^{\frac{22.46 \cdot T}{272.62 + T}}, \tag{2}$$

with air temperature, T, given by NHM-SMAP.

For each location, SNOWPACK simulations were obtained for a 5-year time period between September 2010 and September 2015. SNOWPACK simulations were initiated in September 2010 using a snow profile with a uniform snow layer of 10 m with a density of 500 kg m$^{-3}$ and a temperature corresponding to the average air temperature of the aforementioned time period

as a starting point.

SNOWPACK can be used in different modes, namely 'research' (default), 'Japan' or 'Antarctica'. The various modes use different default settings or parameterizations. Although an 'Antarctica' mode exists in which snow deposition processes are likely to be comparable to the driving processes of snow cover evolution on the GrIS, we nevertheless chose to use the 'research' version of SNOWPACK. This is because the 'Antarctica' version (Groot-Zwaaftink et al., 2013) was calibrated for

a specific time period, i.e. the duration of the Antarctic campaign, and was not developed to be used for a longer time period, i.e. several years.

In the 'research' mode, SNOWPACK estimates the density of new snow during precipitation events based on the current meteorological conditions (Schmucki et al., 2014). However, this new snow density parameterization was obtained using density observations and meteorological conditions from an experimental site above Davos, Switzerland (Eastern Swiss Alps),

and is therefore not suitable for an Arctic environment. Therefore, we ran SNOWPACK simulations using constant new snow densities of 300 kg m$^{-3}$, 325 kg m$^{-3}$ and 350 kg m$^{-3}$. These high values were chosen to reflect that snow precipitation almost always occurs under windy conditions, which leads to a very rapid diminution and a high packing density. This means new snow is added subsequently according to the liquid precipitation amounts provided by NHM-SMAP and the corresponding new snow density. SNOWPACK uses a static threshold of +1.2 °C, below which precipitation falls as snow and above which

it falls as rain.



For each time step, SNOWPACK provides profiles of the optical equivalent diameter – an objective quantification for grain size – which can be used to calculate the SSA ($m^2 kg^{-1}$) according to Gergely et al. (2014).

### 2.4 Field Measurements

Field data were gathered during a field campaign in May 2015 across the GrIS. Stations of the GC-Net (Steffen and Box, 2001) were frequented in order to conduct various snow cover observations. These observations comprised standard manual snow profiles, including density using a density cutter (Proksch et al., 2016), as well as snow temperature profiles. Resistance profiles were measured using the SnowMicroPen (SMP; Schneebeli and Johnson, 1998) and snow samples were taken for on-site analysis of SSA using the IceCube (Gallet et al., 2009). In addition, cast snow samples were taken at Summit and analyzed using micro-computed tomography (micro-CT).

For this study, we focused on density profiles derived from the SMP signal (Proksch et al., 2015) as well as SSA measured with IceCube. A comparison of density observed with the density cutter and derived from the SMP signal is shown in Figure 2. A minimum number of 5 and a maximum number of 80 SMP profiles (see Fig. 8) were taken at the corresponding locations. The depth of the SMP profiles, i.e. down from the surface, ranged between 30 cm and the maximum depth of 200 cm, with a median of 90 cm.

SSA profiles were taken at 6 locations, namely Humboldt, NASA-E, South-Dome, Summit, Swiss Camp and TRV-1. Profiles were taken to a minimum depth of 105 cm and to a maximum depth of 120 cm, depending on the location (median = 115 cm). Only one SSA profile was taken at each corresponding location due to time constraints. Snow samples at Summit were taken down to a depth of about 140 cm and were analyzed using micro-CT with regard to density and SSA.

For the comparison of the modeled density (NHM-SMAP-SNOWPACK) and measured density (SMP), an average density was calculated by first averaging all available SMP profiles for each individual location and then calculating the mean density of the resulting profile. Similarly, a mean SSA was calculated by averaging the available profile at each site. In the following, we refer to SNOWPACK simulations forced with NHM-SMAP reanalysis data as NHM-SNOWPACK

## 3 Results

### 3.1 Comparison of GC-Net and NHM-SMAP

A comparison of the hourly measured (GC-Net) air temperature, wind speed and incoming shortwave radiation with the NHM-SMAP equivalents for the Summit station is presented in Figure 3. In general, the observation and reanalysis data are in fair agreement, with coefficients of determination ($R^2$) being 0.89 for air temperature, 0.49 for wind speed and 0.84 for the incoming shortwave radiation. Intercepts of a simple linear regression for air temperature, wind speed and shortwave radiation were calculated as –1.1, –0.4 and 7.8. The corresponding slopes were 1.2, 0.8 and 1.1, respectively. NHM-SMAP tended to be





too warm with a mean error (ME) and a mean absolute error (MAE) of 5.7°C and 6.0°C, respectively. Measured and modeled values deviated strongly below –35°C. It is possible that NHM-SMAP underestimates atmospheric inversion and the stability of the air. Furthermore, the wind speed tended to be overestimated (ME = 1.2 m s$^{-1}$; MAE = 1.9 m s$^{-1}$) and the incoming shortwave radiation underestimated (ME = –17.9 W m$^{-2}$; MAE = 48.9). The higher modeled wind speed could be a reason for the generally too high air temperature in NHM-SMAP.

### 3.2 NHM-SMAP Reanalysis

A comparison of NHM-SMAP reanalysis data for 14 locations of the GC-Net is presented in Figure 4. Hourly NHM-SMAP 2 m air temperature, wind speed and incoming shortwave radiation from NHM-SMAP are shown. The stations are sorted (left to right) by decreasing latitude. In general, NHM-SMAP 2 m air temperature seems to be homogenous over the GrIS and variances can be explained by the elevation difference between the individual stations. For example, as the highest station, Summit (3,254 m a.s.l.) shows the coldest median temperature compared to Swiss Camp, the lowest station (1,149 m a.s.l.). The NHM-SMAP 2 m wind speed also tends to be homogenous across the GrIS, with a median speed of about 5 m s$^{-2}$. However, the maximum wind speeds tend to increase with decreasing latitude. A similar pattern was found for the incoming shortwave radiation (NHM-SMAP), which also showed increased median and maximum values with decreasing latitude.

The yearly precipitation totals (NHM-SMAP) between July 2012 and July 2015, i.e. over 3 years, are summarized in Table 1. Precipitation totals were calculated from July to June of each year. The greatest yearly precipitation sum was recorded at South Dome (1,145 mm y$^{-1}$) and the lowest sum (69.4 mm y$^{-1}$) at NASA-E. The maximum hourly precipitation amounts occurred at Swiss Camp (25.6 mm) and the least precipitation on an hourly basis was recorded at NASA-E (2.1 mm). The median yearly precipitation across the GrIS was found to be 328 mm y$^{-1}$.

A simple linear regression for yearly precipitation amounts using latitude as a predictor (Fig. 5) shows that almost 75% of the variation in the yearly totals can be explained by the latitude (R$^2$ = 0.74; intercept = 4933; coef. latitude = –61.5). Subsequently adding longitude (R$^2$ = 0.90; intercept = 4424; coef. latitude = –65.1; coef. longitude = 17.1) and elevation (R$^2$ = 0.94; intercept = 5085; coef. latitude = –66.4; coef. longitude = 12.8; coef. elevation = –0.16) as predictors shows that the yearly variations can be explained by the geographical location in 90% and 94%, respectively.

### 3.3 Comparison of Observations and Simulations

#### 3.3.1 Density

The comparison of observed (SMP) and simulated (NHM-SNOWPACK) mean density is shown in Figure 6. The best results in the simulations were obtained using different constant densities for new snow events depending on the geographical location of the station of interest. As is depicted in Figure 1, 300 kg m$^{-3}$ was used for all stations north of Summit, 325 kg m$^{-3}$ for the





Summit and Crawford Point stations, and 350 kg m$^{-3}$ for all remaining stations south of Crawford Point. The MAE was found to be 4 kg m$^{-3}$ and the ME was –1.0 kg m$^{-3}$, indicating a very minor underestimation of the mean density. Although the mean simulated values are in good agreement with the observations, the simulations failed to reproduce the detailed observed stratigraphy (Fig. 8).

### 3.3.2 Specific Surface Area

The SSA was measured using IceCube at 6 locations across the GrIS. The comparison of observed and simulated SSA is shown in Figure 7. The simulation systematically underestimated the observed (IceCube) mean SSA (ME = –6.7 m$^2$ kg$^{-1}$). The mean SSA derived by micro-CT was in good agreement with the simulation. However, as shown in Figure 8, the simulated SSA tended to be underestimated in the upper part (<60 cm) and overestimated in the lower part of the profile (>60 cm). An SSA profile derived from micro-CT measurements is only available for Summit.

### 3.3.3 Accumulation Depth

In July 2012, the GrIS experienced an island-wide surface-melting event forming a melt-freeze crust, which was subsequently buried. During the 2015 field campaign this melt-freeze crust was detected using the SMP at 6 locations across the GrIS. The depth of this melt-freeze crust varied between 1.1 m and 1.8 m (median = 1.5 m). The SMP used can only measure down to a maximum depth of about 2 m, and consequently the melt-freeze crust could only be detected down to that depth.
We compared the observed depths of the melt-freeze crust to the simulated depth (Fig. 9). The depths were found to be in fair agreement with the observations. The MAE was 0.29 m (ME = 0.26), representing a general overestimation of accumulation. The largest deviation from the observation was about 0.7 m, which was found for the NEEM station, i.e. 0.7 m too much snow depth above the 2012 melt-freeze crust.

### 4 Discussion

We compared meteorological reanalysis data (NHM-SMAP), specifically air temperature, wind speed and incoming shortwave radiation, with their corresponding measured (GC-Net) values. Although measurements and reanalysis were found to be in reasonable agreement (Fig. 2), differences remained to a certain extent (see section 3.1). The stations of the GC-Net are subject to extreme conditions of subzero temperatures and wind exposure. Therefore, sensors are sometimes covered with rime and, although an automatic data quality check is applied, a certain unquantifiable error remains. A comparison with high-quality data from Summit station, for example, would allow a better quantification of the errors of the reanalysis data. However, the reanalysis data at Summit was found to be in fair agreement with literature values using high-quality data (e.g. Cullen and Steffen, 2001).



The simulated meteorological conditions – at least for the investigated air temperature, wind speed and incoming shortwave radiation – appear to be rather homogeneous over the GrIS, whereas apparent differences between the stations can be explained by their geographical location, i.e. latitude and longitude, as well as elevation.

Snow cover simulations were conducted with the 1-D physically based snow cover model SOWPACK. For this study, we used SNOWPACK in 'research' mode. This version was originally developed and calibrated for avalanche warning applications in Alpine terrain (Lehning et al., 1999). Nevertheless, SNOWPACK simulations using constant densities for new snow ranging from 300 to 350 kg m$^{-3}$ were found to be in good agreement with the observed (SMP) mean density of the upper snow cover. However, the snow cover stratigraphy could not be reproduced by SNOWPACK, showing a lack of understanding of the fundamental processes of the formation and evolution of snow cover on the GrIS. The formation of sastrugi, snow dunes or hardening of the surface layer due to saltation are local effects. Currently, these effects cannot be modeled with a 1-D snow cover model and are also poorly understood. In fact, the differences found between the measured and reanalysis weather data did not significantly affect the modeled snowpack stratigraphy.

The simulated snow accumulation depth at 6 stations with reference to a buried melt-freeze crust, resulting from the 2012 melt event, was found to be in reasonable agreement with the observations; however, deviations of up to 0.7 m were identified. These deviations could be the result of an underestimation of settling or an overestimation of precipitation – respectively, new snow amounts, or that the 'melt-freeze crust' was formed by preferential infiltration below the July 2012 snow surface. Rapid and deeper infiltration is a well-known phenomenon in subfreezing snow (Tseng et al., 1994).

The SSA was measured using IceCube and a systematic difference of SSA was found when compared with the simulations. Whether this systematic underestimation of about 5 m$^2$ kg$^{-1}$ is a result of an insufficient modeling of the snow microstructure or a measuring error of IceCube remains unknown. However, the analysis of snow samples taken at the same location with regard to the SSA using micro-CT showed a better agreement with the simulations. The reason for the deviation between the SSA derived from micro-CT and IceCube might be that the samples used were very hard and extremely small, broken snow particles artificially increased the reflectivity and, hence, the SSA. However, this point must be further investigated.

## 5 Conclusions

Meteorological reanalysis data (NHM-SMAP) were used to force the snow cover model SNOWPACK at 14 locations (GC-Net) across the GrIS.

NHM-SMAP meteorological parameters, air temperature, wind speed and incoming shortwave radiation were found to be in fair agreement with the observations (GC-Net). The yearly precipitation amounts, maximum wind speed and also the incoming shortwave radiation showed an increasing trend with decreasing latitude. Simulated air temperature and wind speed seem to be homogenous across the GrIS and the average properties can be parameterized by latitude and altitude.

The SNOWPACK simulations were compared to measured data obtained during a field campaign in May 2015. Beside standard manual profiles, the measurements included high-resolution profiles of density and SSA of the upper snow cover



derived from the SMP and IceCube, respectively. A good agreement between the average measured (SMP) and simulated (SNOWPACK) density was found by nudging fixed 'new snow' densities ranging between 300 and 350 kg m$^{-3}$ depending on the geographical location. The simulated SSA showed a systematic offset of about 5 m$^2$ kg$^{-1}$ compared to the IceCube measurements. The simulated SSA at Summit was in good agreement with the measurements derived from micro-CT.

Although the mean density was in good agreement with the simulations, the observed snow cover stratigraphy (density and SSA) was not reproduced by the simulations.

The accumulated depth of snow since the 2012 melt event was found to be in fair agreement with observations, although deviations of up to 0.7 m were observed for some stations (MAE = 0.29 m). However, it is not completely clear if the 'melt layer' represents the surface of the July 2012 snowpack or is a deeper-lying, refrozen, partially water-saturated layer. We

conclude that the simulation of the detailed stratigraphy and evolution of the past and future snowpack of the GrIS is an unsolved problem with the used simulation model.

**Acknowledgements**

We thank Lino Schmid for help in measuring the snow profiles. For fruitful discussions on SNOWPACK modeling the authors would like to thank Dr. Charles Fierz and Dr. Mathias Bavay. The field research was supported by NASA's Cryospheric

Sciences Program, and the Division of Polar Program of the US National Science Foundation. M. Niwano was supported in part by the Japan Society for the Promotion of Science (JSPS), Grant-in-Aid for Scientific Research (A), No. 16H01772 (SIGMA project) and No. 15H01733 (SACURA project), the Global Change Observation Mission - Climate (GCOM-C) / the Second-Generation Global Imager (SGLI) Mission, the Japan Aerospace Exploration Agency (JAXA), the Experimental Research Fund for Global Environment Conservation, the Ministry of the Environment of Japan, and the Grant for Joint

Research Program, the Institute of Low Temperature Science, Hokkaido University.

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





**Tables:**

**Table 1: Summary statistics for NHM-SMAP grid point locations, i.e. latitude, longitude and elevation, as well as simulated precipitation sums per year and maxima between July 2012 and July 2015. The stations are ordered by decreasing latitude, i.e. north to south.**

| ID | Name | Location | | | Precipitation | |
|---|---|---|---|---|---|---|
| | | Latitude | Longitude | Elevation | Sum | Max. |
| | | ° (N) | ° (W) | m | mm y$^{-1}$ | mm |
| 1 | Humboldt | 78.53 | 56.83 | 1995 | 200.4 | 5.7 |
| 2 | Tunu_N | 78.02 | 33.99 | 2113 | 72.5 | 3.7 |
| 3 | NEEM | 77.41 | 51.13 | 2476 | 278.7 | 4.8 |
| 4 | TRV_1 | 77.16 | 48.16 | 2622 | 200.4 | 2.8 |
| 5 | GITS | 77.14 | 61.04 | 1887 | 528.7 | 11.2 |
| 6 | EGRIP | 75.61 | 35.93 | 2718 | 89.1 | 2.1 |
| 7 | NASA_E | 75 | 30 | 2631 | 69.4 | 2.1 |
| 8 | NASA_U | 73.84 | 49.5 | 2369 | 377.4 | 7.4 |
| 9 | Summit | 72.58 | 38.5 | 3254 | 191.8 | 3.1 |
| 10 | Crawford_Point1 | 69.88 | 46.99 | 2022 | 613.4 | 9.3 |
| 11 | Swiss_Camp | 69.57 | 49.32 | 1149 | 940.3 | 25.6 |
| 12 | NASA_SE | 66.48 | 42.5 | 2425 | 936.0 | 9.4 |
| 13 | Saddle | 66 | 44.5 | 2559 | 701.9 | 9.0 |
| 14 | S_Dome | 63.15 | 44.82 | 2922 | 1144.5 | 15.2 |




**Figures**

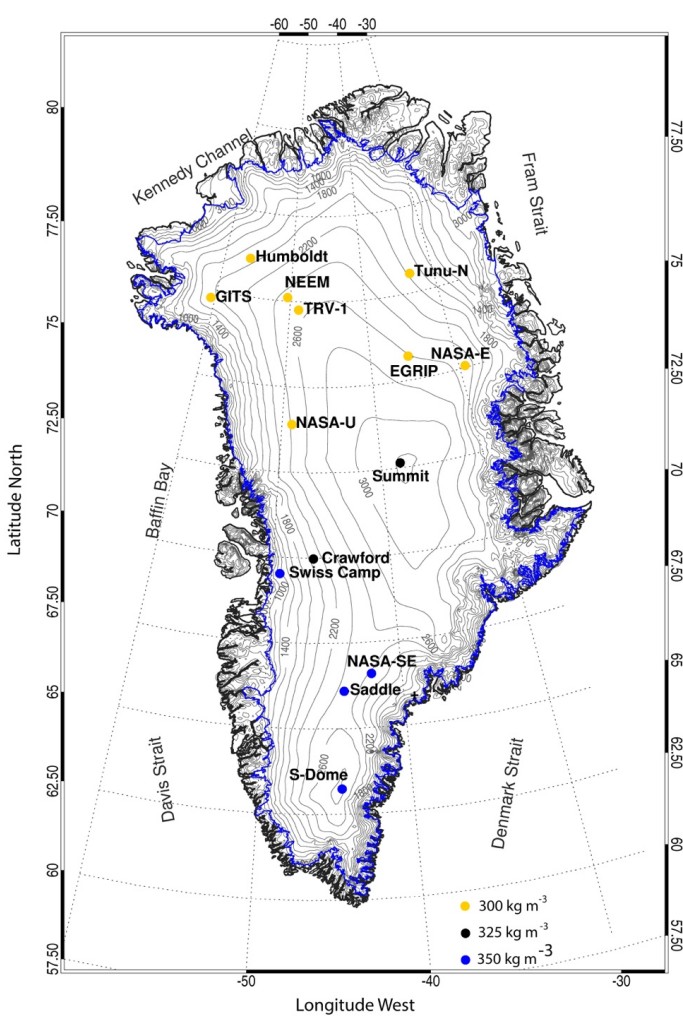

**Figure 1: Map of Greenland showing the locations of the NHM-SMAP grid points. Different colors correspond to different fixed new snow densities as used for NHM-SNOWPACK simulations.**



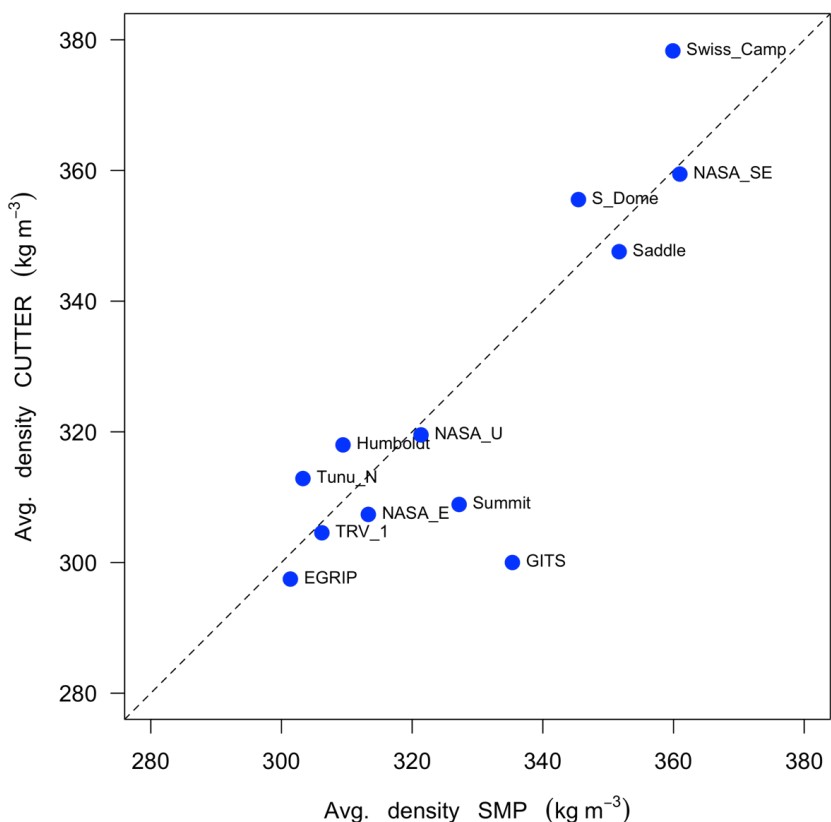

**Figure 2: Comparison of averaged density derived from the SMP signal and measured with the density cutter at 12 locations across the GrIS in May 2015. The dashed line shows the one-to-one relationship.**





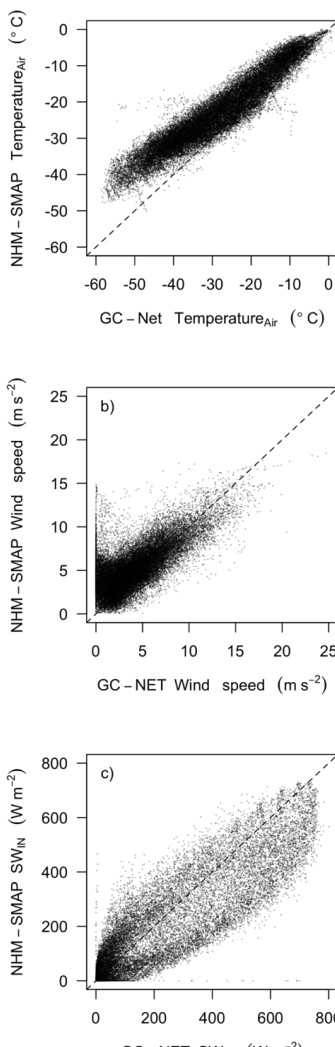

**Figure 3: Comparison of measured (GC-Net) and simulated (NHM-SMAP) (a) air temperature (2 m), (b) wind speed and (c) incoming shortwave radiation for Summit station. The dashed line shows the one-to-one relationship.**





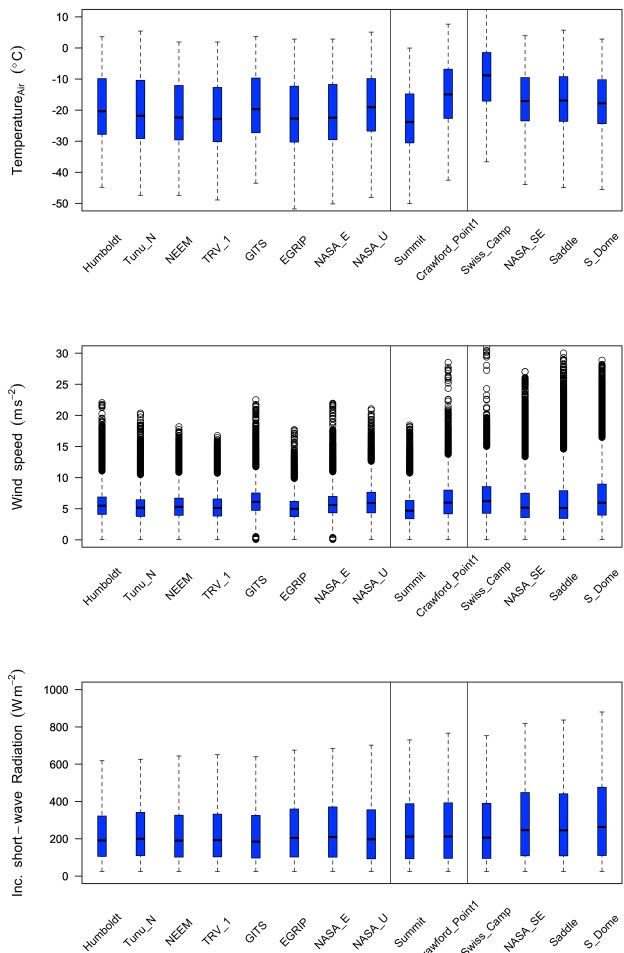

**Figure 4: Boxplots of NHM-SMAP air temperature (top), wind speed (middle) and incoming shortwave radiation (bottom) for 14 stations across the GrIS. Note that stations are sorted by decreasing latitude (left to right). Vertical lines separate grid points where a fixed density of 300 kg m⁻³, 325 kg m⁻³ or 350 kg m⁻³ was used (left to right).**




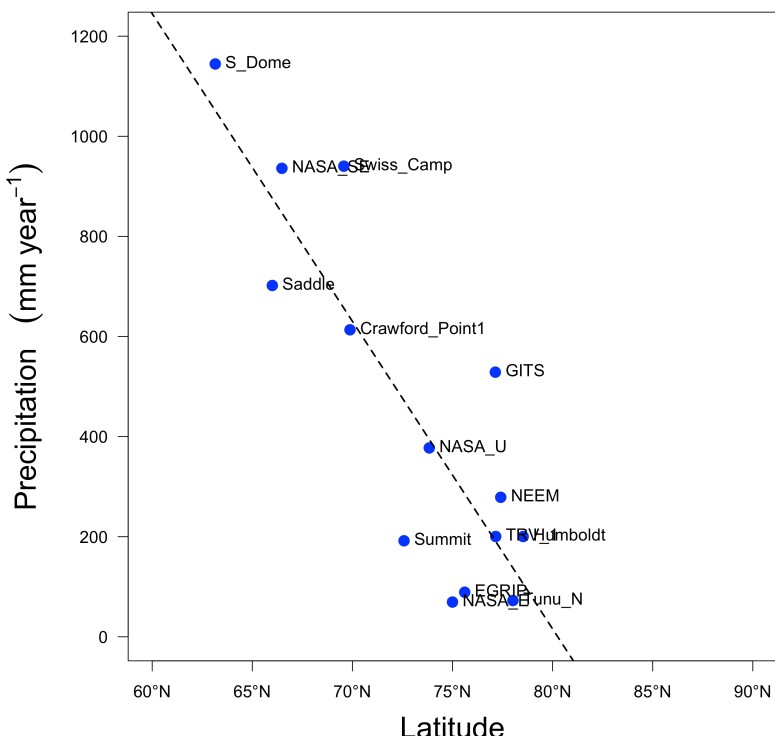

**Figure 5: Yearly precipitation sums from NHM-SMAP between July 2012 and July 2015 with latitude. The dashed line shows the results of a simple linear regression.**





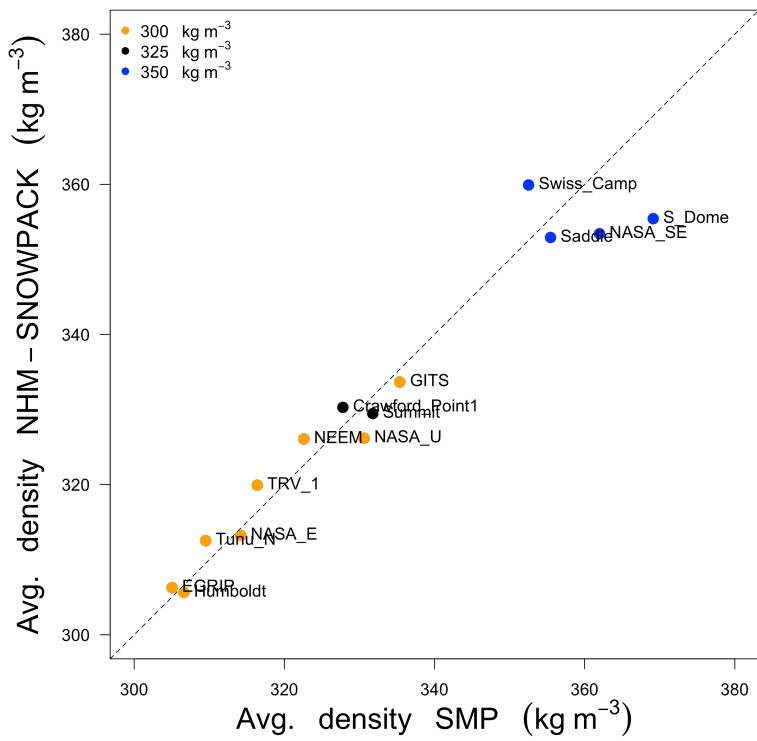

**Figure 6: Comparison of the measured (SMP) and simulated (NHM-SNOWPACK) average density for 14 locations across the GrIS. Colors indicate simulation with different fixed new snow densities of 300 kg m$^{-3}$ (blue), 325 kg m$^{-3}$ (green) and 350 kg m$^{-3}$ (orange). The dashed line shows the one-to-one relationship.**





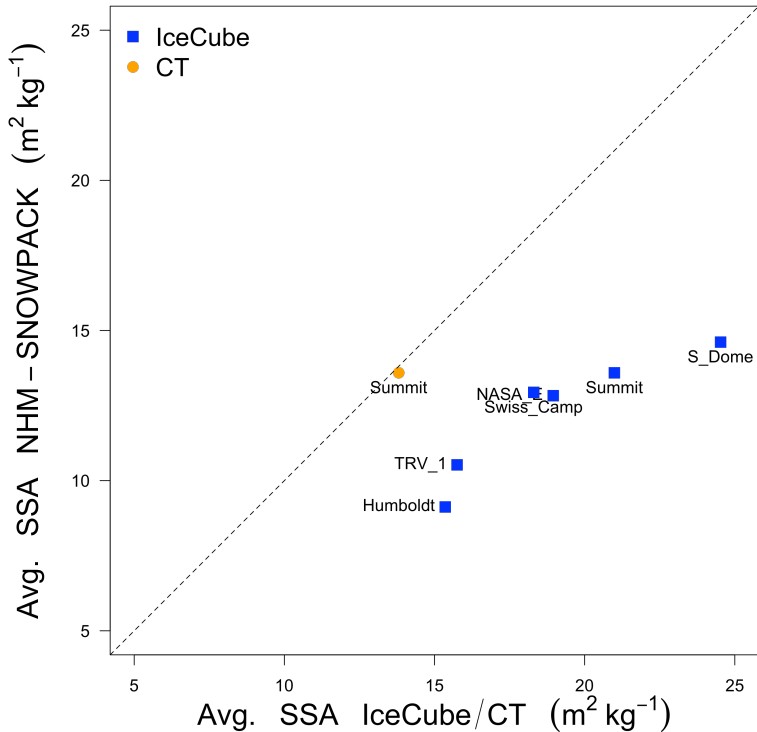

**Figure 7: Comparison of measured (IceCube, blue squares) SSA with the corresponding simulations (NHM-SNOWPACK). The filled circle (orange) shows the SSA derived from snow samples analyzed with CT for a single location (Summit).**





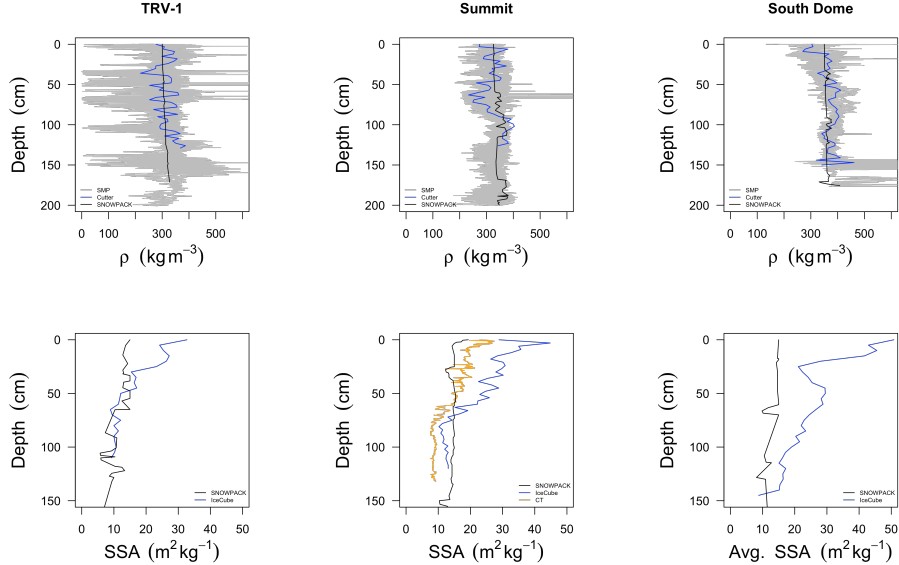

**Figure 8: Comparison of measured density top with the SMP (gray) and the density cutter (blue) as well as a comparison of measured SSA (IceCube, CT; bottom) with corresponding NHM-SNOWPACK simulations (black) for TRV-1, Summit and South Dome, respectively. Note that multiple SMP measurements were taken at the corresponding locations. SSA measured by CT (orange) was only available for Summit.**





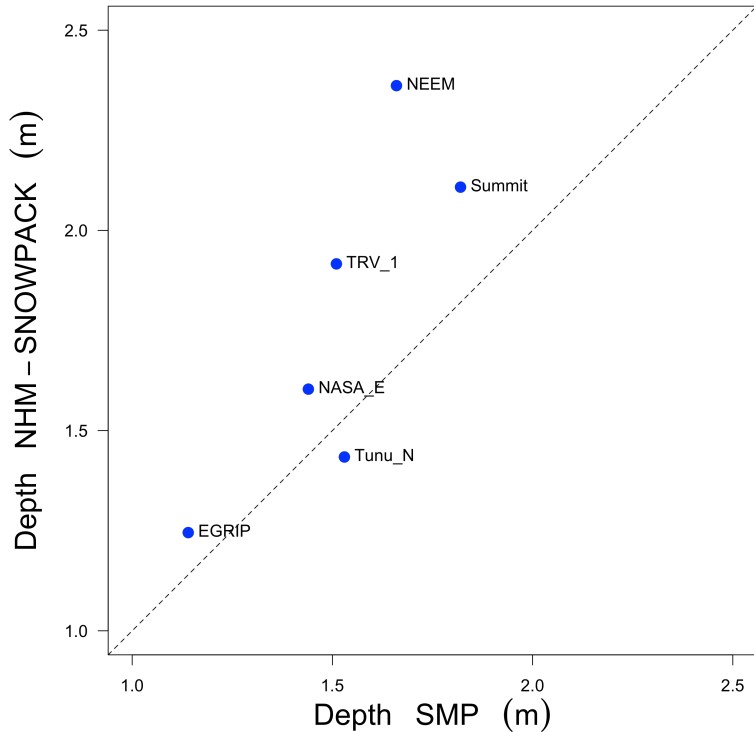

**Figure 9: Comparison of measured (SMP) and simulated depth (NHM-SNOWPACK) of the 2012 melt layer for 6 locations across the GrIS.**