# Peer review of "Measured and Modeled Snow Cover Properties across the Greenland Ice Sheet"

_The Cryosphere, 2017_

## Referee Comment (RC1) · F. Dominé (Referee) · 23 May 2017

**Review of "Measured and Modeled Snow Cover Properties across the Greenland Ice Sheet" by Bellaire et al., submitted to *The Cryosphere.***

This work reports the results and subsequent modeling of a spatially very extensive field campaign across the Greenland ice sheet (GrIS) where snow physical properties were measured in order to understand their spatial and temporal variability. This work may help improve our understanding of the energy budget of the GrIS and ultimately our ability to better predict its contribution to sea level rise.

**General evaluation**

Even though most of the results obtained do not seem to be shown in the paper, I was truly impressed by the breadth of the campaign. A very large number of site were studied, so many that I wonder how the logistics were organized? Did the authors travel on land, giving them the opportunity to observe spatial variations in a continuous manner or did they fly?

There is therefore significant potential in this impressive work. Unfortunately, this potential does not seen adequately exploited here. Most data are not shown and none is available as supplementary material. The modeling work is disappointing. The authors perform what I consider very preliminary simulations where they do not detail key model features. Simulations in general do not reproduce observations and when they do, it is only global averages, often obtained using adjustable parameters so that the actual value of the simulations is not obvious. Sadly, there is no real investigation of why the model does not work. No critical examination of the model physics is performed to try to explain its lack of performance. Furthermore, it is not carefully written. Previous work on the topic is not adequately mentioned and not discussed at all. Crucial details to allow the reader to evaluate the science are often lacking, figure captions are often inadequate, and obviously the paper was not proof-read by all the authors before submission. All in all, this paper looks more like a preliminary campaign report than a true scientific paper. In my opinion, it does not come close to meeting the scientific requirements for publication in The Cryosphere. I am very sorry to come to this conclusion, as I speculate that there should be material of exceptional value for a very valuable publication, but I have to recommend rejection, while hoping that the authors will devote the time this work probably deserves to draft a careful, more complete and elaborate resubmission.

**Major points**

- Experimental data and comparison with previous work. The authors report only a small fraction of their data. How surprising to read a snow physics paper where snow pits were dug without seeing a single stratigraphic profile! How about recapitulative graphs of physical properties? How about a study of trends in physical properties with latitude and altitude? The authors do a nice job correlating precipitation with geography, and in general spend a lot of time on meteorology and reanalysis data, but this is supposed to be (from the title) a snow physics paper. The potential to draw conclusions on spatial variability is just not explored. Regarding temporal variability, the authors briefly mention the seminal work of (Benson, 1996) (actually Benson 1960, but both are in fact equivalent) but do not discuss his

results in comparison with theirs. Likewise, (Carmagnola et al., 2013) report density and SSA profiles obtained at Summit in 2011 but these are not even mentioned by the authors. Additional work by (Alley et al., 1990), (Lyapustin et al., 2009) and perhaps by (Hakala et al., 2014) may also be considered.

- Modeling work. The authors conclude that SNOWPACK fails to simulate the vertical density and SSA profiles of the GrIS, but do not discuss why in depth. I expected an investigation of physical processes potentially responsible for that, in order to possibly improve SNOWPACK. The authors mention wind transport, and this 2-D process cannot be simulated by their 1-D model, but why not envisage an approach similar to that used by (Libois et al., 2014)? In any case, wind transport can only be invoked for properties near the surface. At depth below 5 to 20 cm near the top of the GrIS, this process is inoperative and a 1-D model should work. Is the compaction scheme of SNOWPACK deficient, perhaps because of the low average temperatures? Is the temperature gradient sufficient to drive vapor fluxes that will perturb the density profile, as indicated by (Domine et al., 2016) from observations in the Arctic? The authors did perform temperature profile measurements and there may be buried sensors in a few places (Swiss Camp, Summit?) so they can probably test for that. Regarding SSA, no indication regarding how this is calculated by SNOWPACK is given so that the reader has no idea what is happening. True, this can be found in references, but the golden rule is that a paper can be understood without looking up references.

**Minor points**

1. Does a Li-COR photodiode have the same spectral range as that given by reanalyses? My understanding is that the photodiode measures up to 1100 nm while reanalyses would be over the solar spectrum.
2. Since the density of fresh snow is adjusted in SNOWPACK and since the model does not simulate much variation with depth (i.e. much change after deposition), is it meaningful to claim that the average density is well reproduced by the model? I get the impression that in fact the authors adjust the density on a case-by-case basis so that simulations fit observations and then claim that the model works… Not very convincing.
3. P 5, top. Why not just mention that SSA=6/ice density*diameter? By the way, this relationship was proposed long before (Gergely et al., 2014) so that the choice of this reference is strange.
4. Section 2.4. First paragraph. Figure 1 should be mentioned here. Perhaps mention that samples were transported to Davos for microCT measurements there (I am guessing).
5. P 6, line 3. Replace "air" with "atmosphere". By the way, would there be any impact of the too high wind speeds and winter temperatures given by reanalyses on snow properties?
6. Captions of Figures 6 and 7. Please be more explicit. Mention explicitly that these are averages and specify over what depth.
7. Figure 8. Obviously the SMP data for TRV-1 has a major problem. I can't really imagine the density at 1.5 m depth jumping from 0 to 600 within a few cm. Did all co-authors really

carefully read the paper before submission? Blue can't easily be told from black in the legend. How about red? "Density top" is a strange wording.

**References**

Alley, R. B., Saltzman, E. S., Cuffey, K. M., and Fitzpatrick, J. J.: Summertime formation of depth hoar in central greenland, Geophys. Res. Lett., 17, 2393-2396, 1990.

Benson, C. S.: Stratigraphic studies in the snow and firn of the Greenland ice sheet, U.S. Army Snow, Ice, and Permafrost Research Establishment. Engineer Research and Development Center (U.S.) Research Report 70, 1-93 pp., 1996.

Carmagnola, C. M., Domine, F., Dumont, M., Wright, P., Strellis, B., Bergin, M., Dibb, J., Picard, G., Libois, Q., Arnaud, L., and Morin, S.: Snow spectral albedo at Summit, Greenland: measurements and numerical simulations based on physical and chemical properties of the snowpack, The Cryosphere, 7, 1139-1160, 2013.

Domine, F., Barrere, M., and Sarrazin, D.: Seasonal evolution of the effective thermal conductivity of the snow and the soil in high Arctic herb tundra at Bylot Island, Canada, The Cryosphere, 10, 2573-2588, 2016.

Gergely, M., Wolfsperger, F., and Schneebeli, M.: Simulation and Validation of the InfraSnow: An Instrument to Measure Snow Optically Equivalent Grain Size, IEEE Trans. Geosci. Remote Sens., 52, 4236-4247, 2014.

Hakala, T., Riihela, A., Lahtinen, P., and Peltoniemi, J. I.: Hemispherical-directional reflectance factor measurements of snow on the Greenland Ice Sheet during the Radiation, Snow Characteristics and Albedo at Summit (RASCALS) campaign, Journal of Quantitative Spectroscopy & Radiative Transfer, 146, 280-289, 2014.

Libois, Q., Picard, G., Arnaud, L., Morin, S., and Brun, E.: Modeling the impact of snow drift on the decameter-scale variability of snow properties on the Antarctic Plateau, Journal of Geophysical Research: Atmospheres, doi: 10.1002/2014JD022361, 2014. 2014JD022361, 2014.

Lyapustin, A., Tedesco, M., Wang, Y. J., Aoki, T., Hori, M., and Kokhanovsky, A.: Retrieval of snow grain size over Greenland from MODIS, Remote Sens. Environ., 113, 1976-1987, 2009.

---

## Referee Comment (RC2) · Anonymous Referee #2 · 27 Jul 2017

Synopsis The study 1.) uses GC-Net data to evaluate an NHM-SMAP regional climate model simulation and 2.) uses other field data (density cutter, depth of 2012 melt layer, SMP hardness data, Icecube SSA data) to evaluate SNOWPACK snow model simulations of density. The study concludes bulk density is accurately simulated but detailed stratigraphy is not accurately simulated at the level of detail observed in the field data.

Critique

The study does not spend time evaluating the development of stratigraphy over time despite the fact that GC-Net stations record surface height hourly. Therefore, the work would seem to benefit and deepen in value by evaluating SNOWPACK performance in development of stratigraphy by using selected sequences of snow accumulation (and

erosion) in GC-Net surface height data.

Could SNOWPACK not just be driven by GC-Net data?

The study would be improved by including discussion of results in comparison to those of the following and other relevant studies... Kuipers Munneke, P., S.R.M. Ligtenberg, B.P.Y. Noël, I.M. Howat, J.E. Box, E. Mosley-Thompson, J.R. McConnell, K. Steffen, J.T. Harper, S.B. Das and M.R. van den Broeke, 2015. Elevation change of the Greenland Ice Sheet due to surface mass balance and firn processes, 1960–2014. The Cryosphere, 9:2009-2025.

The use of different new snow densities to tune the model produces better fits but could also mask other error sources or process that the study could reveal.

Throughout, if adjectives like "good" are to be used, they should be accompanied by quantities allowing the reader of the article to judge for themselves model performance. Better would be to greatly reduce the use of adjectives.

The agreement of precipitation vs the simple explanatory factors of latitude and longitude may be useful for the accumulation area but below equilibrium line altitude, the relationship will break down. So, to not over interpret, refer only to the upper accumulation area when making points about the utility of the regression approach.

pg 8 line 21-23 interpretation of SSA vs SSA derived from SMP is speculative and is a point I don't find to be convincing. I think you should suggest other factors that could cause the bias and make some 'further investigations'.

Minor points

A more sophisticated treatment of the rain threshold than +1.2 C seems worthwhile, especially when the modeling is applied at lower elevations where rain is more common.

Equations 1 and 2 are unnecessary. Don't use the abbreviation GrIS, does not save

significant text volume Figure 3 solar irradiance clustering below the dashed line and above 200 W per square m suggests a time offset. Try adjusting the time coordinate of the GC-Net data and I suspect you will find a tighter relationship. Figure 8, for density plots, use square edged line style to more realistically represent the density cutter data regarding solar irradiance, please replace "incoming" with "downward" throughout